# Functional Properties of Meat in Athletes’ Performance and Recovery

**DOI:** 10.3390/ijerph19095145

**Published:** 2022-04-23

**Authors:** Martina di Corcia, Nicola Tartaglia, Rita Polito, Antonio Ambrosi, Gaetana Messina, Vincenzo Cristian Francavilla, Raffaele Ivan Cincione, Antonella della Malva, Maria Giovanna Ciliberti, Agostino Sevi, Giovanni Messina, Marzia Albenzio

**Affiliations:** 1Department of Agriculture, Food, Natural Resources and Engineering (DAFNE), University of Foggia, 71100 Foggia, Italy; martina.dicorcia@unifg.it (M.d.C.); antonella.dellamalva@unifg.it (A.d.M.); maria.ciliberti@unifg.it (M.G.C.); agostino.sevi@unifg.it (A.S.); 2Department of Medical and Surgical Sciences, University of Foggia, 71100 Foggia, Italy; nicola.tartaglia@unifg.it (N.T.); antonio.ambrosi@unifg.it (A.A.); 3Department of Clinical and Experimental Medicine, University of Foggia, 71100 Foggia, Italy; rita.polito@unifg.it (R.P.); ivan.cincione@unifg.it (R.I.C.); 4Department of Translational Medicine, Università degli Studi della Campania “Luigi Vanvitelli”, 80138 Naples, Italy; gaetana.messina@unicampania.it; 5School of Engineering, Architecture, and Motor Sciences, Kore University of Enna, 94100 Enna, Italy; vincenzo.francavilla@unikore.it

**Keywords:** diet, meat, physical activity (PA), health, wellbeing, muscle recovery, exercise-induced muscle damages (EIMDs)

## Abstract

Physical activity (PA) and sport play an essential role in promoting body development and maintaining optimal health status both in the short and long term. Despite the benefits, a long-lasting heavy training can promote several detrimental physiological changes, including transitory immune system malfunction, increased inflammation, and oxidative stress, which manifest as exercise-induced muscle damages (EIMDs). Meat and derived products represent a very good source of bioactive molecules such as proteins, lipids, amino acids, vitamins, and minerals. Bioactive molecules represent dietary compounds that can interact with one or more components of live tissue, resulting in a wide range of possible health consequences such as immune-modulating, antihypertensive, antimicrobial, and antioxidative activities. The health benefits of meat have been well established and have been extensively reviewed elsewhere, although a growing number of studies found a significant positive effect of meat molecules on exercise performance and recovery of muscle function. Based on the limited research, meat could be an effective post-exercise food that results in favorable muscle protein synthesis and metabolic performance.

## 1. Introduction

Since the last decades of the previous millennium, global interest in the issue of health and the concepts of “healthy life” is growing, thus contributing to raise public awareness on the issue of prevention and promotion of “healthy habits”. Physical activity (PA) represents an excellent, practical, and low-cost method of improving health. PA has a wide range of physiological consequences on many body systems [1,2,3,4,5,6,7].

However, despite of the large variety of positive effects of PA, some negative physiological and biomechanics changes occur. They are responsible for the reduction of muscle strength capacity and the appearance of muscle pain, swelling, and stiffness; a symptomatology commonly attributable to exercise-induced muscle damages (EIMDs) [8,9]. Hotfiel et al. [10] postulated that the ultrastructural muscle injury following intense exercise may be due to muscle membrane damage, sarcomere disorganization, protein degradation, autophagy, and local inflammatory response. It has also been suggested that EIMDs are strictly associated with perturbations of electrolyte imbalances, endocrine, and immune systems, resulting in an increased production of reactive oxygen and nitrogen species (RONS) [11] as a likely consequence of inflammatory-mediated recovery processes.

A nutritional intervention was reported to be an effective countermeasure in reducing acute and chronic inflammations and supporting the immune system [12]. It is unlikely that a dietary intervention interacts with the mechanical damage of muscle but rather that it contributes to a modulation of the damage and the subsequent healing process at the level of the oxy-inflammatory cascade in the secondary phase of EIMDs [8]. It has been widely reported that the preventive impact of a diet rich in natural antioxidants outweighs the protective effect of supplementation [13].

Therefore, the utilization of functional foods, rich in bioactive molecules, such as proteins, fatty acids, minerals, and vitamins, has received much attention recently. Bioactive molecules represent dietary compounds that can interact with one or more components of live tissue, resulting in a wide range of possible health consequences such as immune-modulating, antihypertensive, antimicrobial, and antioxidative activities [14].

Athletes consume about 4 portions of meat/week, considering that 35 g of meat provides about 10 g of protein, the meat protein intake is important [15]. Functional activity of meat and meat products is mainly due to taurine, L-carnitine, choline, alpha-lipoic acid, conjugated linoleic acid, glutathione, creatine, coenzyme Q10, minerals (heme irons, zinc), and bioactive peptides produced during post-mortem aging and cooking methods.

PA increases the need for some vitamins and minerals, some of which are highly present in an assimilable form in animal origin products. For example, Zinc intervenes in numerous very important functions such as the growth, construction, and recovery of muscle tissue [16]. Several studies revealed that both male and female athletes suffer from “sports anemia,” which can impair physical performance and is linked to a zinc deficit [17] with meat being the best food source.

For this and other purposes, meat might be useful as an intervention to reduce muscle injury signs and accelerate force recovery.

Therefore, after a summary of current knowledge relating to the EIMD, its causes and consequences; this review provides an integrated overview of the functional properties of meat molecules and the benefits of meat consumption in athletes’ nutrition.

## 2. The Role of Nutrition in Physical Activity

PA is a systematic sequence of physical, psychological, and tactical preparatory events [18] aimed at achieving maximum performance in a race or maintaining optimal health status.

The achievement of good athletic performance is the result of an optimization process of body functions and specific adaptation to physical effort [19]. The resolution of the adaptation process is a phenomenon that implies the restoration of a physiological equilibrium.

Considering all the factors that can influence athletic performance, such as genetics, talent, and the type of training; good nutrition is certainly the simplest to manage.

A healthy and balanced diet is essential for people who practice PA, as it offers adequate calories and nutrients to fulfill energy and nutritional demands. Good nutrition, able to satisfy the higher energy demand, helps the athlete in performance and recover quickly.

Macronutrients such as carbohydrates, and fats represent the primary energy sources; however, proteins play a key role in muscle growth and recovery. Protein and essential amino acids ingestion can stimulate muscle protein synthesis and help in the healing of injured tissues before, during, and especially after the PA. Furthermore, micronutrients such as minerals and vitamins, whose requirement increases significantly during the PA, are involved in metabolic processes, oxygen transfer and delivery, and tissue repair; while antioxidants can support adequate immune function and protect against oxidative damage exercise-induced [20,21,22].

It should therefore be emphasized that proper nutrition is a decisive factor because: (1) promotes the growth and development of the organism, (2) supplies useful energy for a performance, and (3) help with the synthesis and recovery of muscle tissue from EIMDs [23]. However, it should be considered that the entity of the EIMDs can be influenced by the intensity, type, and duration of the exercise and therefore nutrition can be remodeled according to these functions [24]. Generally, in athletes’ diet most of the daily caloric intake (55–65%) must be represented by carbohydrates, proteins must account for 10–15% of total calories (preferably a combination of proteins from foods of animal and vegetable origin), while the remaining 25–30% of the caloric intake must come from fats which, if the physical performance is long-lasting and of low intensity, are used as an energy source [15].

However, as mentioned above, the daily requirement of a sportsman varies according to the type of sport and the metabolism involved (aerobic-anaerobic) as reported in Table 1.

Aerobic activities (marathon, cross-country and middle distance, skiers, cyclists), in fact, require a large supply of carbohydrates capable of guaranteeing a sufficient supply of glycogen to provide energy during prolonged efforts.

On the contrary, in anaerobic activities (weightlifting, shot put, hammer or discus) the protein intake is important, which favors the development of muscle mass.

The Mediterranean diet is unquestionably one of the greatest eating models to follow today [26], alongside it, new dietary trends such as the ketogenic diet [27], the Intermittent Fasting diet [28], or the alkaline diet have found widespread diffusion. Due to the restrictive regimes proposed by the aforementioned dietary plans, their implementation must be dependent on the type of physical activity performed. For example, the reduction in caloric intake associate to the Intermitted Fasting might result in a decrease in lean body mass and strength, and therefore, potentially, a decrease in the capacity to conduct resistance exercise [29,30].

It’s widely accepted that an inadequate intake of energy and hydration substances can result in decreased strength and endurance performance, increased soreness, and the consequent risk of muscle damage. Therefore, for athletes, a healthy and balanced diet must not only be a nutritional priority to meet energy demands, but above all a fulfillment of physiological needs thus ensure optimal performance during exercise.

## 3. The Physiology of Muscle Recovery

It is widely assumed that EIMDs arise when a person is repeatedly subjected to long-lasting heavy training characterized by an eccentric muscle contraction. EIMDs represent the main cause of the reduction of muscle strength capacity and consequently of sports performance [8,9].

Hody, et al. [31] proposed an extensive review on the mechanisms involved in the onset of EIMDs. The complex mechanisms of EIMDs could be simplified in a two-phases damage model, the initial phase starts during the exercise when excessive tension and repeated eccentric contractions lead to a greater fiber, sarcoplasmic reticulum, and membrane injury; subsequently, the release of calcium into the cytoplasm contributes to a secondary myofibrillar damage in skeletal muscle [8,9,10,11,12,13,14,15,16,17,18,19,20,21,22,23,24,25,26,27,28,29,30,31,32].

Indeed, calcium is responsible for the activation of muscle proteases, it triggers a chain reaction resulting in mitochondrial dysfunction, activation of cell death signaling, and secretion of stress hormones (i.e., catecholamines, cortisol) [31,33].

As a result, immediately following exercise, an increase of leukocytes, primarily neutrophils and later macrophages, has been widely reported [31,34]. With neutrophil mobilization in damaged muscle, an oxidative-inflammatory cascade occurs; necrotic fibers are degraded, and proteases, cytotoxic enzymes and RONS are produced [35].

Specifically, the overproduction of RONS stimulates the Keap1-Nrf2 (Kelch-like ECH-associated protein 1-nuclear factor erythroid 2-related factor 2), signaling pathway which regulates the gene expression of several antioxidants and anti-inflammatory molecules [36] and represents the main regulator system of cytoprotective responses to oxidative stress [37,38].

The occurrence of oxy-inflammatory mechanisms as a consequence of training, by activating muscle stem cells known as satellite cells [39], is therefore the basis of organism adaptations and contributes to muscle recovery and remodeling [40,41].

Since the activation of satellite cells provides the muscle with the potential to regenerate following damaging PA; boosting satellite cell activity and extending their activation may be physiological mechanisms on which dietary intervention might be used to modulate muscle recovery and performance.

## 4. Meat Molecules in Muscle Recovery

Meat and meat products are key components of a healthy and balanced diet. They are concentrated natural sources of high-quality proteins as well as other nutrients such as fats, vitamins (B2, B6, B12, pantothenic acid and niacin) and minerals (iron, zinc, phosphorus, and selenium) that exert a beneficial role on growth, preservation, and repair of the human body.

Furthermore, meat is rich in bioactive molecules and endogenous antioxidants with well-established beneficial properties linked to their anti-inflammatory, immunoregulatory, and oxidative stress protective activity [42,43]. Once absorbed by the ingestion of animal origin food, these molecules can affect the structure and function of organs and tissues by acting as biomolecule substrates [44].

The nutritional profile of meat (on average consisting of 75% water, 19% protein, 2.5% fat, 1.2% carbohydrates and 1.65% nitrogen compounds [45]) is heavily influenced by the species, feeding regimen, cut and cooking techniques utilized [46].

For example, the content of L-carnitine in beef steak and lamb meat ranges between 64.6–78.6 and 190 mg/100 g fresh weight, but chicken breast only reaches values of 13–34.4 mg/100 g fresh weight [47]. Table 2 provides an overview of the content of the main biologically active molecules in meat cut from different animal species.

### 4.1. Proteins and Amino Acids

It is well accepted [61] the role of meat proteins avoids catabolism and stimulates muscle growth. On average, 100 g of meat, provides up to 20 g of useful protein to meet human daily needs [62]. This is impressive considering that the amount of protein necessary to enhance muscle and albumin protein synthesis after a single session of endurance exercise is around 20–25 g of high-quality intact protein [63].

Aside from quantity, the quality of the proteins is critical in increasing muscle protein growth. Meat proteins include well-balanced necessary amino acids that aid in damaged tissue development and repair [64,65,66]. For example, due to its immunomodulatory activity, glutamine is recognized as playing a key role in muscle recovery [67,68]. Glutamine is found in various food [69], however the best way to get it is by consuming meat and other animal products. Nakhostin-Roohi et al. [70] demonstrated that taking 1.5 g/kg/day glutamine for 7 days can lower the level of creatine kinase, an intramuscular biomarker enzyme of EIMDs. Furthermore, glutamine was found to be effective in increasing the levels of glutathione, a tripeptide of great importance in the prevention of sports injuries [71,72].

Particularly relevant in EIMDs prevention is the role of branched chain amino acid (BCAA) [73,74]; for which it was observed [75] that an intake of 200 mg/kg/day for at least 10 days can be helpful to prevent moderate muscle damage. Among all, leucine, the main BCAA in meat, was proven to be involved in the activation of skeletal muscle protein synthesis [76].

Moreover, Ra et al. [77] showed in a 2013 study that combining BCAA and taurine, compared to supplementing a single compound, might be an effective method for reducing several biomarkers of EIMDs.

### 4.2. Taurine and L-Carnitine

Unlike supplements, meat could be considered an effective strategy against EIMDs thanks to the balanced and complete supply of amino acids especially of functional ones (taurine, and L-carnitine) with well-defined physiological roles in anti-oxidative and anti-inflammatory reactions, as well as neurological, immunological, and cardiovascular function [78].

These compounds are abundant in all beef cuts; indeed 30 g of dry beef provides enough taurine to fulfill the daily physiological demands of a healthy 70 kg adult person [78].

The role of these molecules in muscle recovery has been extensively reviewed over the past few years. Correspondingly, Thirupathi et al. [79] and Sawicka et al. [80] discussed the link between taurine and L-carnitine with the modulation of exercise-induced RONS formation.

Taurine and L-carnitine are endogenous molecules that are typically synthesized in the body; however, when cells are under physiological stress, this quantity is insufficient [81,82].

The amino acid intake from meat consumption represents an effective strategy in maintaining the concentrations of L-carnitine and taurine at the cellular level, especially considering that glutathione (GSH) and taurine share the same precursor, and this could increase the oxidative damage induced by muscles stressful conditions [79].

According to Thirupathi et al. [79], taurine could prevent the susceptibility at subsequent pathology PA-induced by protecting cell membranes against oxidative stress and controlling inflammation [83]. L-carnitine, instead, is involved in the maintenance of mitochondrial function and metabolic processes concerning the synthesis of branched-chain amino acids, the generation of energy, and the induction of β-oxidation processes; but it is also recognized as having a transport, thermogenesis, and antioxidant function [50].

### 4.3. Carnosine

A central role in recovery following EIMDs is recognized to both carnosine and, its methylated metabolite, anserine: they are the most abundant antioxidants in meat [84]. Carnosine concentration in meat ranges from 500 mg/kg of the chicken thigh to 2700 mg/kg of pork shoulder [85] instead anserine is more abundant in chicken muscle [86]. Furthermore, the consumption of meat promotes the synthesis of carnosine in the human body thanks to the contribution of β–alanine, a limiting factor in carnosine biosynthesis [87,88].

Markus et al. [89] synthesized the physiological role of carnosine in enhancing exercise recovery. Varanoske and colleagues [90], found a greater reduction of fatigue when compared to the carnosine levels in recreationally trained women [89]. Thus, higher intramuscular carnosine levels were associated with greater fatigue relief, and women who consumed the most dietary protein were likewise more prone to have the highest intramuscular carnosine content.

### 4.4. Creatine

Compared to other foods, meat is also an excellent source of creatine, containing in beef 5,25 mg/g on average, according to Mateescu et al. [91]. Creatine effectiveness, in terms of EIMDs effects amelioration, has been widely demonstrated in several studies [91,92]. Inflammation, oxidative stress, calcium homeostasis, changes in glycogen storage, and satellite cell activity in damaged muscle are all plausible strategies creatine-associated for preventing EIMDs [92,93].

According to Jiaming and Rahimi [92] meta-analysis’ findings, creatine reduces creatine kinase (CK) and lactate dehydrogenase (LDH) concentrations immediately after 24–48–72–96 h of exercise. CK and LDH have been widely used as indicators of muscle micro-damages and, therefore, are often considered indirect biomarkers of EIMDs [94].

### 4.5. Glutathione (GSH)

As stated above, glutathione demonstrated [95] to be useful in reducing muscular fatigue. In fact, intense exercise results in a reduction in plasma and tissue concentrations of GSH as well as an increase in the oxidized form compared to the reduced one [96,97]. The human body, on the other hand, has been found very effective in absorbing GSH from the diet [98]. According to Aoi et al. [95], including GSH in athletes’ diets increased GSH plasma concentrations and led to an improvement of antioxidant defenses and aerobic metabolism performance.

### 4.6. Conjugated Linoleic Acid (CLA)

Besides the protein fraction, conjugated linoleic acid (CLA) from the meats’ lipid fraction showed an anti-fatigue effect in athletes [99].

Meats, especially red ones, have been demonized for years due to their saturated fat content. Meats contain not only saturated fat but a large amount of monounsaturated fatty acid (MUFA) and polyunsaturated fatty acid (PUFA) with proven beneficial effects [100].

CLA, among all, are important for athlete general health [22]. Kim et al. [101] reviewed the impact of CLA on muscle metabolism, specifically observed, in several animal trials, an improvement in the exercise outcome with CLA treatment, however, human studies are still limited and require further investigation.

In addition to its high amino acids content, meat represents an important source of micronutrients such as minerals, vitamins, and alpha-lipoic acid.

### 4.7. Iron

Physical activity is associated with a constant demand for minerals, particularly in endurance or long-lasting sports, where the loss is significant, due to sweat and other factors. Iron is a necessary component of hemoglobin and myoglobin proteins, which are present in red blood cells and muscles, respectively. During exercise, hemoglobin and myoglobin transport oxygen to the tissues, and athletes’ performance greatly depends on the efficiency of this system. Sim et al. [102] widely summarized the role of iron in athlete performance. Fatigue is the main indicator of a possible iron deficiency [103]; a phenomenon to which women are particularly susceptible [104]. Iron deficiencies can be avoided by the intake of highly absorbable and usable iron, such as that contained in beef. However, dietary intake alone may not be enough to meet the higher daily iron requirements of active and trained women and therefore, supplementation may be necessary for some circumstances [104].

### 4.8. Coenzyme Q10 (CoQ10)

The meat’s nutritional composition is characterized by a high concentration of group B vitamins (thiamine, riboflavin, vitamin B6, niacin, pantothenic acid, biotin), they are involved in the energy conversion during PA, while folate and vitamin B12 are necessary for red blood cells and protein synthesis, tissue repair, and maintenance [43,44,45,46,47,48,49,50,51,52,53,54,55,56,57,58,59,60,61,62,63,64,65,66,67,68,69,70,71,72,73,74,75,76,77,78,79,80,81,82,83,84,85,86,87,88,89,90,91,92,93,94,95,96,97,98,99,100,101,102,103,104,105].

The group B vitamins, besides assuming a fundamental role in human nutrition, have been proven to enhance coenzyme Q10 production (CoQ10) [106]. Meat, poultry, and fish are all excellent sources of dietary CoQ10 [107].

CoQ10, also known as ubiquinone, is a fat-soluble vitamin-like compound found in all cellular membranes [43]. Borekova et al. [60] summarized the main activities of this molecule, among all, the antioxidant properties exceed in terms of both quantity and efficiency that of recognized antioxidants. Therefore, the ingestion of CoQ10 has been observed to be effective in countering the increase in RONS and in modulating the inflammatory signaling [108,109] associated with PA. Additionally, a variety of factors are responsible for CoQ10 deficiency [110]. In sports, greater energy demands may lead to the alteration in mitochondrial metabolism characterized by decreased oxidative phosphorylation and increased physical fatigue. According to Haas [111], CoQ10 ingestion can improve energy metabolism by lowering phospholipase A2 and restoring phosphocreatine levels (a source of physical energy).

### 4.9. Alpha-Lipoic Acid (ALA)

Finally, different clinical trials demonstrated the antioxidants and anti-inflammatory effects of alpha-lipoic acid (ALA) in several diseases. ALA has been used to regenerate the other antioxidants, enhance glucose metabolism, and reduce EIMDs in a variety of tissues [112]. Isenmann et al. [113] found possible beneficial effects of ALA in maintaining performance and reducing muscle damage, in line with earlier studies of Morawin et al. [114], who observed significantly lower CK values after eccentric exercise when athlete received ALA supplementation.

Many of the functions described here and summarized in Table 3 have shown an impact of meat molecules on red-ox processes.

However, Margaritelis et al. [118] emphasize the critical role of oxidative stress in exercise adaptations. EIMD is a complex phenomenon, if on the one hand oxidative stress and inflammatory phenomena can increase muscle damage, on the other hand they allow the restoration of functions and adaptation thus contributing to recovery.

The alternative definition of “antioxidant paradox” [119] recognize antioxidants as inhibitors of the signaling effects of exercise induced RONS production and thus an obstacle to the mechanism of muscle adaptation and recovery.

Based on these evidence, nutritional strategies should be adopted with pragmatism. The use of the great majority of the molecules of animal origin in food formulations and as supplements in the prevention and treatment of various PA diseases should be discouraged in favor of a balanced diet which should always be considered as the main nutritional intervention strategy.

## 5. Importance of Meat Bioactive Molecules in Enhance Athletes’ Health and Performance

Currently, due to the growing popularity of plant-based diets, many people believe that eating meat is just harmful to health. However, giving up this essential source of bioactive molecules might really cause more damage than benefits.

Considering the tendency towards the abuse of foods of animal origin by developed Western populations and the relative negative health implications that it entails, the adoption of a more vegetarian nutritional style is today considered a directly and indirectly preventive attitude towards various pathologies and uncomfortable conditions, such as the famous “diseases of well-being”. This is because vegetarian diets bring lower levels of saturated fat, cholesterol and animal proteins indirectly related to health problems, as it is not the proteins themselves that are harmful, but the food processing methods. Vegetarian diets offer higher levels of carbohydrates, fiber, magnesium, potassium, folic acid and various types of antioxidants such as vitamin C, vitamin E, polyphenols. Data literature reported a lower risk of ischemic heart disease and cancer, but no effect on overall mortality or cerebrovascular disease in vegetarian subjects [120]. A further analysis combining two large studies found that vegetarians in the UK have an all-cause mortality like that of carnivores [120]. The positions on the beneficial or harmful effects of the vegetarian diet are controversial. In relation to sport, the athlete needs more energy which derives mainly from carbohydrates and, to a lesser extent, from lipids; more amino acids: they derive from proteins and have numerous functions. The essential ones (AAE) are especially important, necessary for the protein synthesis of structural proteins of the muscle, enzymes involved in energy processes; in addition, more minerals and water involved in sweating: especially magnesium and potassium, although their quantity varies according to the extent of sweating itself. More vitamin and mineral antioxidants such as vitamin C, vitamin E, zinc and selenium. An increase in oxidative processes inexorably determines an increase in the quantity of free radicals [121]. More precursors and enzymatic components such as the B vitamins and certain minerals. Enzymes are biological catalysts essential to cellular processes which, in sports, increase exponentially [122]. However, the vegetarian athlete also suffers from a lack of some fundamental macro and micronutrients. The intake of proteins and especially of AAE in vegetarian and especially vegan diets tends to be lower than that in omnivorous diets [123,124]. Creatine is an amino acid molecule used as a reserve of phosphate groups by muscle cells [125]. The body can synthesize it and there is no form of nutritional deficiency related to it. In vegans, internal production is maximal and some believe it is not enough. The integration of creatine in these subjects who practice strength sports, even more so if they are vegan, can increase strength performance. Supplementing with creatine is therefore recommended for vegan strength athletes and bodybuilders who follow the same diet. Despite the richness in total lipids and lipophilic molecules (such as vitamin A and vitamin C) of Western vegetarian diets, there is a lack of the two semi-essential omega 3 fatty acids: eicosapentaenoic acid (EPA) and docosahexaenoic acid (DHA). Moreover, in the vegetarian athlete there is a lack of vitamin B12, Iron, Calcium, Zinc and Phosphorus, essential microelements for the regulation of fatty acid metabolism, Calcium is essential for muscle contraction and all have a strong anti-oxidative power [120,121,122,123,124].

In the sportsman’s diet, meat has the dual purpose of meeting energy needs and providing for the protection, repair, and growth of tissues.

Meat, although not recognized as the main energy source, contains some molecules capable of improving athletic performance, especially during intense but short-term efforts. Creatine, for example, contributes to increased phosphocreatine energy stores [126] and, thus, the capacity to perform high-intensity activities. Some B vitamins (thiamine, riboflavin, vitamin B6, niacin, pantothenic acid, biotin) are also involved in energy conversion during exercise, while folate and vitamin B12 are required for blood cell production, protein synthesis, tissue repair, and maintenance [105]. Proteins, the main and essential meat nutrient, are associated with the improvement of several biological activities. They are in-volved in the growth and recovery of various muscle groups, bones, and other organs. All the essential amino acids that athletes require to rebuild muscle tissue and perform at their best are found in meat proteins. The amount of protein required increases with age and varies based on the type of PA and the desired goal [61]. Besides protein, the potential health benefit of meat can be attributed to many bioactive molecules with proven antioxidant, anti-inflammatory, and immunomodulatory properties. Taurine, BCAAs, L-carnitine, creatine, CLA, and ALA can stimulate the immunological response in skeletal muscle, whereas iron and vitamin-derived compounds such as CoQ10 have been proved, by the attenuation of interleukin-6 (IL-6) and C-reactive proteins levels [109] in blood, to strongly support immune functions. Potential antioxidant properties are also associated with some meat molecules. Several studies have shown the role of CoQ10, ALA, and glutathione in the preservation of prooxidant/antioxidant homeostasis, in forming a complex with metal ions, and regenerating enzymes involved in RONS scavenging [43,95]. In conclusion, it is worth noting that meat, when consumed in the right proportion, provides the body with several bioactive molecules that assist athletes to stay healthy while also improving their performance and muscle repair [Figure 1].

It is not impossible to achieve good results in athletic performance while following a vegetarian or vegan diet. However, dietary choices must be made based on the load and the type of training, for this reason, the omnivorous diet is obviously the most complete one, with the right amount of macro and micronutrients, without absolutely demonizing meat, which like as previously reported, it has many beneficial properties both in a protein and biomolecule intake.

## 6. Conclusions

Nutrition also plays a fundamental role in sport as it can influence athletic performance or improve recovery after sport. Nutrition is therefore a key factor for sports performance, contributing to the athlete’s success and determining an improvement in body composition and an increase in physical/cognitive performance in view of a sporting event.

Physical activity, athletic performance and recovery from exercise are influenced and improved by a correct diet with a view to an adequate and balanced intake of macro and micronutrients. An athlete’s diet load should be like that recommended for the general population with energy intake divided into ap-proximately 55% carbohydrates, approximately 15% protein and approximately 30% lipids. This food ration must be divided into 3–4 main meals, and of course it can be modified in terms of macronutrients according to the type of sport, and also to the training period. In this scenario, it is important to say that the protein share is fundamental in an athlete, as proteins play a fundamental role in post-exercise functional recovery. In younger subjects, proteins ensure good muscle hypertrophy, which is why protein intake is generally slightly improved, especially among young athletes. Current data suggest that the amount of dietary protein needed to support metabolic adaptation, repair, remodeling, and protein turnover generally ranges from 1.2 to 2.0 g/kg body weight per day [62]. Higher intakes may be indicated in short periods during intensive training. However, in the event of an energy restriction or sudden inactivity that could occur as a result of an injury, increasing your protein intake to 2.0 g/kg per day or higher could be helpful in preventing the loss of lean mass. For these reasons, the meat consumed is a valid strategy to meet the athlete’s protein requirements. Meat consumed in the correct proportions in a healthy and balanced diet is an important food intervention to support the health and performance of athletes. Due to its compositional complexity, offer a balanced supply of nutrients and functional compounds. The molecules believed to be responsible for its functional activity have found wide application as food supplements however they should be used with caution and with specific target [9]. Molecules intake through meat allow to circumventing problems related to the specific dosage and timing of intake of food supplements. Amino acids deriving from meat proteins digestion, following absorption into the bloodstream, can interact with cells of tissue, with muscle cells, modulating biochemical and physiological cell pathways, influencing athletic performance and recovery. Certainly, protein and meat consumed plays a key role in the athlete’s diet, but further studies are useful to clarify the molecular pathways of this action.

## Figures and Tables

**Figure 1 ijerph-19-05145-f001:**
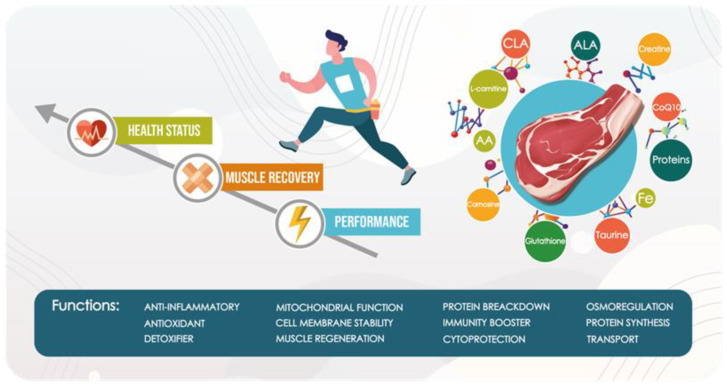
Schematic representation of the functional activities of meat bioactive molecules.

**Table 1 ijerph-19-05145-t001:** Caloric demands per hour in different sport activity [25].

**Speed Running**	500 cal/h
**Throwing**	460 cal/h
**Jumping**	400 cal/h
**Speed swimming**	700 cal/h
**Tennis (single)**	800 cal/h
**Wrestling**	900 cal/h
**Boxing**	600 cal/h
**Basketball**	500 cal/h
**Football**	400 cal/h

**Table 2 ijerph-19-05145-t002:** Contents of major biologically active meat molecules (mg/100 g) in raw meat cuts of different animal species.

Bioactive Molecules	Contents of Major Biologically Active Meat Molecules (mg/100 g)						References
	Beef	Veal	Pork	Lamb	Chicken	Turkey	
**Taurine**	43.1	39.8	61.2	43.8	17.8 (fillet)	29.5 (fillet)	[48]
**L-carnitine**	64.6–78.6 (muscle) 226 (ribs) 10.7 (liver)	78.2 (shoulder) 132.8 (sirloin) 6.5 (liver)	21.1 (shoulder) 17.7 (leg) 40.2 (ribs) 10.7 (liver)	190	13–34.4 (fillet) 69.2 (liver)	21.2–200	[47,49,50]
**Carnosine**	375 (loin)	-	313 (loin) 449 (ham)	39.3 (shoulder)	180 (breast) 63 (thigh)	66 (wings)	[51]
**Creatine**	401 (muscle) 298 (heart) 16 (liver)	488 ± 41	247–374 (ham)	278–511	482 ± 44	284 ± 62	[52,53]
**Glutathione (GSH)**	13.4	23.9 (cutlet) *	23.6	23.9 (cutlet) *	13.1 (fried) * 8.7 (roasted) *	8.7 (fried) *	[54]
**Conjugated linoleic acid (CLA)**	2.9–4.3	2.7	0.9	5.6	0.7–1.5	2.0–2.5	[55,56]
**Choline**	330 (liver) 46.1–56.3 (lean meat) 100 (neck)	310 (liver) 70.1 (loin, tenderloin) 61.6 (shoulder)	69 (ground) 79 (neck)	54.5 (chops) 75.8 (leg) 35.4 (shoulder, arm)	94.3	220 (liver) 130 (heart)	[57,58]
**Total Iron**	2.07 ± 0.1 (sirloin) 2.35 ± 0.2 (fillet)	0.85 ± 0.3 (fillet)	0.36 ± 0.1 (loin) 0.49 ± 0.1 (chump chop)	2.23 ± 0.4 (chop)	0.63 ± 0.2 (wing) 0.70 ± 0.1 (leg thigh) 0.63 ± 0.1 (leg lower part) 0.40 ± 0.1 (breast)	0.50 ± 0.1 (breast) 0.88 ± 0.2 (leg lower part) 0.99 ± 0.3 (leg thigh)	[59]
**Coenzyme Q10 (CoQ10)**	3.65 11.3 (heart) 3.9 (liver)	-	2 (ham) 12.6 (heart) 2.27 (liver)	-	1.4	-	[60]
**Alpha-lipoic acid (ALA)**	0.06–0.11 (liver) 0.07–0.10 (heart)	0.01–0.02 (muscle) 0.03–0.05 (liver) 0.05–0.07 (heart)	0.02–0.03 (ground) 0.02–0.04 (neck) 0.06–0.08 (liver) 0.11–0.16 (heart)	0.02–0.04 (muscle) 0.07–0.08 (liver) 0.05–0.07 (heart)	-	-	[52]

* values referred to cooked meat.

**Table 3 ijerph-19-05145-t003:** Major biologically active meat molecules and their functions in preventing EIMDs.

Bioactive Molecules	Bioactivities Observed	References
**Proteins and amino acids**	(1) Regulatory activity of muscle mass and recovery; (2) Stimulating activity of amino acid and muscle protein synthesis; (3) Anti-inflammatory activity; (4) Antioxidant activity; (5) Modulatory activity of cyclin-dependent kinase 2 (cdk2) gene expression and of proliferating cell activation; (6) Preventive role against myostatin and myogenin mRNA decrease	[67,68,70,73,74,75,76]
**Taurine**	(1) Antioxidant activity; (2) Anti-inflammatory activity	[78,79,83]
**L-carnitine**	(1) Antioxidant activity; (2) Anti-inflammatory activity, (3) Regulatory activity of mitochondrial function and metabolic processes concerning the synthesis of BCAAs, the generation of energy, and the induction of β-oxidation processes; (3) Thermogenesis	[50,78,80]
**Carnosine**	(1) Antioxidant activity; (2) Ion-chelating activity; (3) Anti-glycating activity which also prevents the formation of advanced lipid oxidation end-products	[89]
**Creatine**	(1) Antioxidant activity; (2) Anti-inflammatory activity; (3) Gene transcription of amino acid pools co-regulatory activity; (4) Promote synthesis of myofibrillar protein; (5) Attenuation of plasma muscle damage markers; (6) Regulatory activity of calcium homeostasis; (7) Regulatory activity of energy metabolism; (8) Improve muscle glycogen accumulation	[92,93,115,116]
**Glutathione (GSH)**	(1) Antioxidant activity and detoxification functions; (2) Improve aerobic energy metabolism and muscle contraction maintenance	[95]
**Conjugated linoleic acid (CLA**	(1) Promote bone formation and muscle mass growth; (2) Improve exercise outcome by modulating testosterone; (3) Antioxidant activity; (4) Anti-inflammatory activity; (5) Attenuation of plasma muscle damage markers; (6) Modulatory activity of skeletal muscle metabolism	[22,99,101]
**Iron**	(1) Modulate inflammatory/iron regulatory hormone (hepcidin) responses; (2) Regulate iron-dependent metabolic pathways; (3) Promote the adaptation to hypoxic environments	[102]
**Coenzyme Q10**	(1) Antioxidant activity, regenerate other antioxidants; (2) Modulatory activity of energy metabolism; (3) Modulatory activity of inflammatory signaling; (3) Pro-angiogenic effect; (4) Improve oxygen supply	[60,108,109,117]
**Alpha-lipoic acid (ALA)**	(1) Antioxidant activity; (2) Anti-inflammatory activity; (3) Attenuation of plasma muscle damage markers	[112,113]

## Data Availability

Data is contained within the article. Authors can use this data for research purposes only by citing our research article.

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
