# Peer review of "Functional Properties of Meat in Athletes’ Performance and Recovery"

_ijerph, 2022, doi:10.3390/ijerph19095145_

Round 1

Reviewer 1 Report

Review - Functional properties of meat in athletes’ performance and recovery

Re-reviewing the manuscript

Specific issues:

The authors made major revisions to the manuscript. Unfortunately I still have a lot of comments.

  1. The bibliography still has errors.
  2. I am thinking about using the manuscript (item 4) in the implementation of the topic.
  3. A large part of the introduction describes the importance of physical activity - please focus unnecessarily on the topic of nutrition.
  4. Lines 174-176, 183-192 diverge from the main topic
  5. Adequate description to the topic of the work begins from line 199.
  6. Please rewrite the table no. 3 in its present form is illegible
  7. Please see the manuscript abbreviation policy, use equally throughout the manuscript
  8. The conclusions have been changed. Please refer to the vegetarian diet used by athletes. Is it effective or harmful?

Author Response

Dear Review, 

in attached our response. 

thank you 

Reviewer 2 Report

This review provides an integrated overview of the functional properties of meat molecules and the benefits of meat consumption in athletes' nutrition.

Major concerns

As a reviewer, I was interested in reading this information, and I found some interesting facts. I'm reviewing this paper for the second time. However, no major changes have been made. My recommendations would be based on the following. It would be appropriate for authors to specify the objective of the study. It is necessary to write the Methods and specify how authors collected the information and used it in the systematic analysis of the literature review. In the next step, I would recommend to write the Results. After all that, I would suggest describing the Discussion section before writing the Conclusions.

Lines 427-457: The conclusions contain information that is already known.

Lines 457-463: The conclusions contain information that is not accurate: „Molecules intake through meat allow to circumventing problems related to the specific dosage and timing of intake of food supplements. Meat protein can interact with cells of tissue, in particular in the muscle cells modulating biochemical and physiological cell pathways, influencing athletic performance and recovery. Certainly, protein and meat consumed plays a key role in the athlete's diet, but further studies are useful to clarify the molecular pathways of this action“.

My questions would be as follows:

What problems related with dietary supplements the authors mean?

Can meat proteins really interact with tissue cells? Thus, it is known that when any rich in protein food is consumed, these proteins in the digestive system are broken down to monomers (in a specific case to amino acids), which are then absorbed into the bloodstream. Meat proteins per se really can't modulate biochemical and physiological cell pathways.

Minor conserns

Lines: 442-443: “Younger proteins ensure good muscle hypertrophy, which is why protein intake is generally slightly improved, especially among athletes“

My question would be as follows:

What does the "younger proteins" mean?

King Regards

Author Response

(The authors gave the same response as above.)

Round 2

Reviewer 1 Report

Thank you to the authors of the manuscript. Manuscript has been corrected

Author Response

We would like again to thank the Reviewer for all valuable comments and suggestions which helped us to improve the quality of the review.

Reviewer 2 Report

Thank you for the opportunity to review an interesting topic „Functional properties of meat in athletes’ performance and recovery“.

In accordance with the traditional narrative review, I would recommend that Authors carry out the good quality systematic review.

The Authors gave just a description of the meat and its composition. However, this information is known and nothing new has been found. The findings do not reflect anything innovative. The majority of athletes are likely to consume meat and meat (sub) products. However, both fish, eggs and dairy products can be the sources of protein and essential amino acids too. I would therefore recommend that athletes' diets be taken into account in an integrated way when assessing recommendations for protein intake.

It is not only the analysis of the composition of the meat that could potentially ensure the relevance of the manuscript. In this context, a comparison of vegetarian diet with a full diet is significantly more important.

Author Response

In attached the response to comments. 

This manuscript is a resubmission of an earlier submission. The following is a list of the peer review reports and author responses from that submission.

Round 1

Reviewer 1 Report

The authors of the paper “Proteins and amino acids, branched chain amino acid (BCAA)” are exploring an interesting topic. However, it seems necessary to structure the article and make a clear distinction between the Methods, Results and Discussion subsections. It seems necessary to improve Table 1. Proteins and amino acids, branched chain amino acid (BCAA) are the same nutrients.

The authors of the manuscript wrote about food supplements in subsections 4.1 and 4.2. The manuscript per se is not focused on taking dietary supplements according to the subject and purpose. Therefore, it is necessary to write only about the amounts of nutrients contained in food, specifically meat.

It should be noted that the effectiveness of dietary supplements has been extensively investigated worldwide. There are many of the consensuses have been published in International journals. Therefore, the information provided by the authors should be adjusted in lines 318 to 320 of the manuscript.

Most of all, the article lacks information on different types of meat and its nutritional composition. It seems necessary to look at sources providing additional information. It is also important to note that there are two types of physical loads: aerobic and anaerobic. The authors' own conclusions should therefore also focus on the different sports and the impact of meat consumption. On the other hand, the conclusions were not based on the results of the study at all, nor the methods of the investigation were clear. The manuscript is written in a poorly way. There are a lot of inaccuracies.

Additionally, the authors should note that during intense exercise, free radical or reactive oxygen and nitrogen species (RONS) production increases and may inhibit muscular contractile function leading to muscle fatigue and performance impairment. Lower doses of RONS appear to be beneficial for training adaptations during acute performance. Although antioxidants play an important role in the protection from RONS, evidence suggests that antioxidant supplementation may prevent the induction of peroxisome proliferator activated receptor-c coactivator (PGC-1α), mitochondrial biogenesis and impair exercise training adaptations.

Reviewer 2 Report

Review - Functional properties of meat in athletes’ performance and recovery.

The idea of the work is interesting. However, due to serious methodological errors, I recommend- reject the manuscript.

Specific issues:

  1. The title of manuscript indicates dietary recommendations (meat consumption by athletes.) The author did not include any food consumption guidelines depending on the sport discipline. The recommendations are general. And the work is a partial description of the importance of individual ingredients in a diet.
  2. The topic related to the physiological and biochemical aspects of the training and biological regeneration process has been completely omitted
  3. Lack of reference to the use of alternative diets by athletes.
  4. Methodically, the work is not a literature review. There is no mention of the Search Strategy for the selection of publications
  5. Conclusion- completely does not match the content of the work. It contains information on telemedicine and even SARS CoV2 infection
  6. Literature inconsistent with the requirements for authors